# Si Photonics FMCW LiDAR Chip with Solid-State Beam Steering by Interleaved Coaxial Optical Phased Array

**DOI:** 10.3390/mi14051001

**Published:** 2023-05-05

**Authors:** Yufang Lei, Lingxuan Zhang, Zhiyuan Yu, Yulong Xue, Yangming Ren, Xiaochen Sun

**Affiliations:** 1State Key Laboratory of Transient Optics and Photonics, Xi’an Institute of Optics and Precision Mechanics, Chinese Academy of Sciences, Xi’an 710119, China; 2University of Chinese Academy of Sciences, Beijing 100049, China

**Keywords:** LiDAR, optical phased array, integrated optics

## Abstract

LiDAR has attracted increasing attention because of its strong anti-interference ability and high resolution. Traditional LiDAR systems rely on discrete components and face the challenges of high cost, large volume, and complex construction. Photonic integration technology can solve these problems and achieve high integration, compact dimension, and low-cost on-chip LiDAR solutions. A solid-state frequency-modulated continuous-wave LiDAR based on a silicon photonic chip is proposed and demonstrated. Two sets of optical phased array antennas are integrated on an optical chip to form a transmitter–receiver interleaved coaxial all-solid-state coherent optical system which provides high power efficiency, in principle, compared with a coaxial optical system using a 2 × 2 beam splitter. The solid-state scanning on the chip is realized by optical phased array without a mechanical structure. A 32-channel transmitter–receiver interleaved coaxial all-solid-state FMCW LiDAR chip design is demonstrated. The measured beam width is 0.4° × 0.8°, and the grating lobe suppression ratio is 6 dB. Preliminary FMCW ranging of multiple targets scanned by OPA was performed. The photonic integrated chip is fabricated on a CMOS-compatible silicon photonics platform, providing a steady path to the commercialization of low-cost on-chip solid-state FMCW LiDAR.

## 1. Introduction

Due to the short wavelength, narrow beam, and good direction of laser, LiDAR has the advantages of a high resolution and strong anti-interference. LiDAR has been widely used in autonomous vehicles [1], robotics [2], aerial mapping [3], atmospheric measurement [4], and augmented reality [5]. Especially with the boom in the autonomous vehicle industry, the research into vehicle LiDAR has received the attention of many research institutions at home and abroad.

LiDAR usually uses beam steering or flash illumination schemes [6] to map the surrounding environment and realize three-dimensional imaging. However, the requirement of laser power for a flash scheme is very high, which brings a great burden to the power dissipation and heat dissipation of the system. In addition, laser power is limited by human eye safety, so the detection range of flash schemes is usually very limited. Commonly adopted beam steering schemes include the mechanical movement of optics, micro-electromechanical systems (MEMS) mirror scanning, and optical phased array beam steering. At present, most commercial LiDAR systems rely on mechanical movement for beam steering. For example, most of DJI’s products use a dual-prism structure; Hesai and Tudar use polyhedral rotating mirrors. The mechanical scanning technology is mature and can realize large angle steering. However, it has limitations regarding the scanning rate, long-term reliability, and system cost. Therefore, solid-state beam steering has attracted wide attention in recent years [7]. Optical phased array (OPA) is a type of solid-state beam steering technology which promises agile and precise optical beam steering free of any mechanical movement.

In addition to the flash illumination and beam steering module, another important core of the LiDAR system is the choice of detection schemes. At present, most commercial LiDAR systems use direct time of flight (dTOF) technology for ranging. The LiDAR emits a beam of pulsed light. The distance between the LiDAR and a target can be obtained by directly measuring the time delay of the received pulses. The ranging accuracy of dTOF is limited by the measurement accuracy of the time, the pulse bandwidth, and the response speed of the photodetector. dToF is also known to be susceptible to interference from ambient light and light from another LiDAR operating in the same wavelength. Compared with dTOF, frequency-modulated continuous-wave (FMCW) is a technology promising high accuracy and immunity to ambient or multi-LiDAR interference thanks to the coherent detection nature by mixing the frequency-chirped reflected signal with a local oscillator to extract the beat frequency corresponding to distances. In addition, the local oscillator acts as an optical amplifier which drastically reduces the demand for output optical power, benefiting power consumption and eye safety. In addition, FMCW LiDAR can not only provide distance measurement, but simultaneously obtain velocity information from the Doppler effect. It is believed that constructing such a location-based distance–velocity 4-dimensional field may benefit a perception system in tasks such as object detection and semantic segmentation. FMCW LiDAR generally adopts a coaxial optical system, where the transmitting and receiving systems share the same optical path, which ensures that the propagation wavefront of the received light matches with the local light, so that the transmitted light and the receiving light are completely coherent. Compared with the design of transmitting and receiving separation, this design simplifies the system design, eliminates the short-range blind area, and is more conducive to multi-channel fusion and improves the frequency of the ranging points. Therefore, the coaxial FMCW ranging technology is also considered as an important development direction of LiDAR [8].

The advancement in integrated photonics provides a predictable implementation path for low-cost on-chip solid-state beam steering LiDAR. In recent years, integrated optical phased array (OPA) has been widely studied because of its solid-state beam control ability [9]. With the rapid development in silicon photonics, on-chip large-scale OPA becomes possible [10]. Optical phased array can also achieve a large scanning angle and high scanning speed [11,12,13]. At the same time, combined with FMCW ranging technology, solid-state FMCW LiDAR chips have been realized [14]. All these research results provide strong support for the development of on-chip solid-state LiDAR.

However, since a low loss circulator cannot be integrated on the chip at present, an on-chip coaxial coherent detection system can be built by a 2 × 2 beam splitter, as shown in Figure 1a. The 2 × 2 beam splitter will cause an inherent loss of 6 dB. The linear frequency-modulated laser enters the 2 × 2 beam splitter along the input light path and is divided into two beams, where one beam is launched into free space along the emission light path, and the other beam is generally consumed or partially used as local light. In addition, the return light reflected by the target in free space is also divided into two beams by the 2 × 2 splitter, where one beam is transmitted along the laser input light path and, finally, lost by the isolator of the laser, and the other beam interferes with the local light to generate a detectable signal. Half of the transmitted and received optical power is divided by the 2 × 2 optical splitter, and the system directly loses 6 dB of optical power. The proposed chip-based LiDAR uses two interleaved optical phased arrays for coaxial transmitting and receiving. The approach saves 50% of the transmitting power compared with a common method using a 2 × 2 beam splitter for such a coaxial configuration as shown in Figure 1a. 

## 2. Design of on-Chip Solid-State FMCW System

The FMCW LiDAR system in this design, as shown in Figure 1b, mainly consists of several parts: a signal generator, tunable laser, transmitting and receiving interleaved coaxial optical phased array (Tx/Rx OPA), interference system, balanced photodetector (BPD), transimpedance amplifier (TIA), and signal processor. The interleaved coaxial optical phased array is added to the FMCW ranging system as a transceiver to realize on-chip solid-state beam scanning. This design saves space and eliminates close-range blind areas. In addition, it can solve the alignment problem of noncoaxial systems. In this design, the working wavelength of the all-solid-state FMCW LiDAR on the chip is 1550 nm because wavelengths around 1550 nm are commonly used in telecom devices and the cost of continuous-wave sources is very low. The band also has the advantages of low atmospheric absorption and eye safety at a high power. It is also simple to perform a slight optical frequency chirp of a DFB laser through injection current modulation. In addition, 1550 nm is the transparent window in silicon. The use of silicon is desirable because of its CMOS compatibility. The theoretical basis of using triangular wave frequency modulation to achieve distance measurement is shown in Figure 1b. The optical path difference between the received signal (Rx) and the local oscillator (LO) is transformed into a low-frequency signal by the method of coherent detection. The relationship between the distance and beat frequency is $f_{b}=\frac{2B}{T}\times \tau =\frac{4Bn_{0}r}{Tc}$ where *τ* is the delay between the received signal (Rx) and the local oscillator (LO), *B* is the laser modulation bandwidth, *T* is the laser modulation period, R is the distance to the target, and *c* is the speed of light.

The traditional system relies on discrete components; facing the problems of large volume and complex structure, the integrated optical chips can be used to achieve a higher integration, smaller size, and lower-cost LiDAR scheme. The transmitting and receiving system, interference beat system, and photoelectric detection system are integrated on the silicon optical chip. The specific design of the optical chip is shown in Figure 2a, including the edge-to-input coupler, directional coupler, optical splitter which consists of a cascade 1 × 2 multimode interference beam splitter, phase shifter array, coaxial transmit and receive grating array, interferer (multimode interference beam splitter, with splitter ratio of 2:2 is used, which is represented by M22), and balanced photodetector. The linear frequency modulation light coupled into the chip is divided into 2 beams by a directional coupler, where 90% of the light is divided into 32 channels by a 5-stage 1 × 2 multimode interference beam splitter network. After adjustment by the phase shifter array, the beam is transmitted from the transmitting array to generate a single-direction transmitting beam, which is reflected by the target and received by the receiving array to generate the signal light. In addition, 10% of the light divided by the directional coupler is taken as the local oscillator. The local oscillator interferes with the signal light on M22 and generates the beat signal. A variable optical attenuator is used to adjust the intensity of the local oscillator and increase the signal-to-noise ratio of the beat signal. The beat signal is detected on the balance detector.

In order to achieve a higher signal intensity, the receiving direction of the receiving array can be adjusted by the phase-shift array, and the intensity of the reference light can be adjusted by the on-chip variable optical attenuator. The system adopts a staggered coaxial transceiver optical phased array design. Compared with a common method using a 2 × 2 beam splitter for such a coaxial configuration, as shown in Figure 1a, this design eliminates the 3 dB loss of the transmitting and saves 50% of the input optical power, in principle, with the same performance. Figure 2b shows an optical micrograph of the chip. Figure 2c shows the actual picture of the chip placed on a one yuan coin, and the corresponding overall size of the chip is 7.1 mm × 2 mm. The photonic chip (Figure 2c) is fabricated on an SOI (silicon-on-insulator) platform with 2 µm buried oxide and 220 nm top silicon by Advanced Micro Foundry (AMF) (Singapore). As shown in Figure 2d, the optical fiber block is pasted on the substrate to realize optical packaging. Electrical packaging is carried out at the same time. There are a total of 70 electrical packaging channels, 64 channels of which are used to control the direction of the phased array of transmission and reception, 3 electrical channels are used for the signal, offset, and grounding of the balance detector, 1 channel is used to adjust the adjustable attenuator to achieve the reference light intensity change, 1 electrical channel is used to monitor the coupling efficiency, and 1 electrical channel is grounded. 

The transceiver optical phased array adopts an interleaved coaxial design to improve the receiving efficiency, and introduces nonuniformity or aperiodicity in the large spacing antenna array to suppress the grating lobes. As the divergence angle of the light beam is inversely proportional to the dimensions of the optical antennas, the long diffraction grating is designed to implement a small divergence angle. The design of the coaxial array with interleaved receiving and transmitting is shown in Figure 3a.

The interleaved coaxial antenna is made of silicon waveguide and periodic nano-blocks. Light will form an evanescent field around the waveguide, and the nano-blocks periodically perturb the evanescent fields to form a weak-radiating grating, leading to a long effective radiation length required for highly directive optical phased arrays. The silicon waveguide is designed for a narrow waveguide, which makes the evanescent field larger. A trapezoidal mode size converter is used to connect the ordinary waveguide and the narrow waveguide. The design parameters of the antenna mainly include the period number of the periodic nano-blocks, period length w1, duty cycle w2/w1, the etching depth of the periodic nano-blocks, the gap between the nano-blocks and the waveguide, the width of the nano-blocks L1, the width of the waveguide wg1, the gap between the transmitting waveguide and the receiving waveguide gap1, the bending radius R, etc. The period of the nano-blocks determines the deflection angle of the antenna. The emitter–receiver efficiency is determined by the etching depth, duty cycle, grating width, antenna interval, and antenna period number. In addition, the number of periods of the periodic nano-blocks also determines the antenna size.

The specific design parameters are shown in Figure 3c (red is the silicon structure, white is the silicon dioxide), where the narrowed waveguide width wg1 is 0.4 μm, the ordinary waveguide width wg is 0.5 μm, and the length of the mode size converter is 4 μm. The period length w1 is 1 μm, the duty ratio is 0.5, the etching depth of the periodic nano-blocks is 220 nm, the gap between the nano-blocks and the waveguide is 0.2 μm, the width of the nano-blocks L1 is 2 μm, the gap between the receiving and transmitting waveguides gap1 is 2.4 μm, the bending radius R is 2 μm, the number of periods N is 350, and the corresponding grating length L is 350 μm. With the increase in the number of periods N, the forward efficiency T of the antenna gradually decreases, and the relationship between T and N is an exponential function with base e, where N is the independent variable and the dissipation coefficient is α. Because the grating is too long to simulate, in order to quantitatively analyze the relationship between the forward efficiency T and the number of periods N at λ0 = 1.55 μm, the value of α is calculated according to the simulation results of FDTD. The average dissipation coefficient α ≈ −0.0093 is obtained by calculating the forward efficiency T of the grating from the 5th to the 40th period, and it is extended to the calculation formula of the forward efficiency T of any period gratings. Then, it is calculated that the forward efficiency of the grating tail is 0.93% when the number of periods is N = 350, and a trapezoidal waveguide is designed to dissipate the transmittance of the tail. The simulated far field of the grating (N = 50) is shown in Figure 4a. The far-field deflection angle corresponding to the grating is 27°, and the diffraction envelope defined as the full width at half maximum (FWHM) of the far field is estimated to be 20°. The beam scanning within the antenna diffraction envelope can be realized by adjusting the phase of the phased array.

Within the diffraction envelope, the steering capability of the OPA is further limited by grating lobes resulting from the high order interference of an array of antennas. A nonuniform OPA has been successfully introduced to suppress the grating lobes. To achieve a higher grating lobe rejection ratio, we used a deep learning-based genetic optimization, as reported in our previous work [15], to simultaneously optimize the antenna locations while meeting the required minimal spacing constraints. The key to the above algorithm is using a deep neutral networks (DNN) model to replace the crossover and mutation operations of the genetic algorithm while the DNN’s weights parameter is also updated in each evolution cycle. In this paper, the optimization problem is represented by two sets of parameters. One represents the locations of all the antennas, and the other represents the far-field optical performance. We use the peak-to-sidelobe suppressing ratio as the figure of merit (FoM). A 1 × 32 1D optical phased array is used in this design, where the minimum antenna spacing is 12 μm and the average antenna spacing is 16μm. The simulated one-dimensional far-field distribution obtained after the optimization of 1 × 32 OPA is shown in Figure 4b. The far field is free of high order grating lobes with a peak-to-sidelobes suppressing ratio of 8 dB, defined as the ratio of the peak to the maximum of any sidelobes in the entire far-field space. In addition, the beam width is 0.6°, defined as null-to-null of the main lobe.

## 3. Results

### 3.1. Result of Optical Phased Arrays

The beam steering test system is composed of an infrared camera and projection screen. An infrared camera is used to obtain the far-field distribution of the optical phased array on the projection screen. The far-field distribution of the optical phased array was controlled by adjusting the voltages, thus changing the phases of the antennas. A typical far-field pattern of an uncontrolled OPA chip is shown in Figure 5a. The far-field intensity distribution is similar to that of a single antenna. The main lobe can be scanned along the θx direction by adjusting the phase shifter, where the phase shifters are independently driven by an electric circuit board composed of digital-to-analog converters (DACs) with a 10 V output swing to provide more than a 2π phase shift. The modified rotating electric field vector (REV) method mentioned in our previous article [16] was used to calibrate the phase and the main lobe scan in the θx direction, as shown in Figure 5a. The calibrated far-field distributions steered to 6 angles (−3°, −1.1°, 0.88°, 2.57°, 4.36°, 8.6°) along the θx axis are shown in overlap in Figure 5b, and the diffraction envelope is shown by the red line in the figure. Figure 5c shows the corresponding far-field distribution in the θy direction, when steering in the θx direction. For a different steering angle, the main lobe is affected by the diffraction of the single antenna, and the intensity of the main lobe varies along the diffraction envelope. In addition, the peak-to-sidelobes suppressing ratio maintains more than 3 dB for all the steering angles, and the scanning range size is about 15°. The maximum grating lobe rejection ratio obtained in the experiment is about 6 dB, which is about 2 dB less than the simulated value, and the beam width of the main lobe corresponding to the maximum grating lobe rejection ratio is 0.4° × 0.8° on the θx axis and θy axis, respectively. The beam width of the main lobe did not change during scanning. 

### 3.2. Result of FMCW LiDAR Chip with Solid-State Beam Steering

The on-chip solid-state FMCW LiDAR testing system is shown in Figure 6. The single-lobe beam and its beam angle of the optical phased array on the LiDAR chip were controlled by adjusting the voltages, thus changing the phases of the antennas. The beam forming of the transmitting OPA was obtained first followed by maximizing the received power of the receiving OPA to ensure the coincidence of the beam angle of both arrays. The mixed photocurrent signal from the balanced photodetector was amplified by the transimpedance amplifier and read out by an acquisition card. Fourier transform was performed on the obtained signal to obtain the beat frequency and the corresponding distance was calculated. In this test, the central wavelength of the laser is 1550 nm, the frequency modulation bandwidth is 5 GHz, the modulation period is 15 KHz, the optical power is 100 mw, and the reflectance of the target is 90%. The range error is mainly affected by the modulation bandwidth. The ranging accuracy is about 1.5 cm. Three targets with different distances and angles are shown in Figure 7a, where the target is a highly reflective cooperative target. The voltage of the transmitting and receiving OPA is controlled synchronously to make the beam deflect to the position of the three targets and acquire the corresponding signals of different targets. The spectrum signals corresponding to different targets are shown in Figure 7b. The peak frequencies are 0.195 MHz, 0.25 MHz, and 0.31 MHz, respectively, and the corresponding test distances of the signals are 18 cm, 24 cm, and 30 cm, respectively, which are consistent with the actual distance results. There is slight crosstalk between the receiving array and the transmitting array, part of the light is directly coupled from the transmitting array into the receiving array, so there are low-frequency signals present. Moreover, the signal frequency value of the target is always large due to the long distance of the actual test. Therefore, we filter the low-frequency signals when processing the data, which will not have too much influence on the ranging. 

A preliminary beam steering performance of the optical phased array was tested by using three targets placed at different angles. The correct distances were obtained when the OPA output beam was steered in the direction of these targets. The experiment serves as a proof-of-concept for a solid-state, lens-free, and chip-integrated LiDAR design.

## 4. Conclusions

In this paper, we present a method to design an all-solid-state FMCW on-chip LiDAR with an interlacing coaxial transceiver array. The transmitter–receiver interleaved coaxial optical phased array is used as the transceiver unit and is added to the on-chip FMCW system. In theory, the proposed chip-based LiDAR uses two interleaved optical phased arrays and saves 50% of the transmitting power compared with the method of a 2 × 2 beam splitter for such a coaxial configuration, as shown in Figure 1a. In addition, due to the solid-state scanning performance of the optical phased array, an on-chip all-solid-state LiDAR system without a mechanical structure and lens can be realized. An on-chip FMCW LiDAR system with a 32-channel transmitter–receiver interleaved coaxial OPA with a beam width of 0.4° × 0.8° and a grid–lobe rejection ratio of 6 dB is demonstrated. The optical phased array used in this reported LiDAR chip has only 32 channels which limits the receiving aperture and, thus, the detection range. We believe that the ranging performance of the LiDAR will be improved with the increase in the number of antennas. The increase in the aperture also reduces the line width of the main lobe and increases the suppression of the grating lobes. The presented work in this paper provides an on-chip lens-free truly solid-state beam steering design which greatly benefits the LiDAR miniaturization and manufacturing complexity. We believe that the coaxial optical phased array concept is scalable and provides a promising way to enable chip-scale-integrated LiDAR products.

## Figures and Tables

**Figure 1 micromachines-14-01001-f001:**
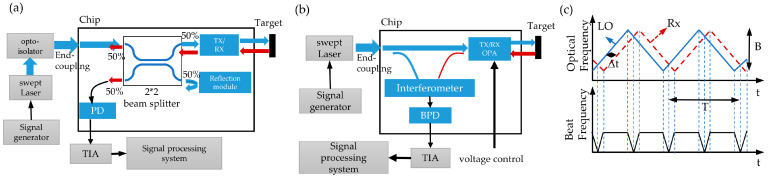
(**a**) Schematic diagram of on-chip coaxial FMCW LiDAR system built by 2 × 2 beam splitter; (**b**) schematic diagram of on-chip FMCW LiDAR system with solid-state beam steering by interleaved coaxial optical phased array; (**c**) ranging principle.

**Figure 2 micromachines-14-01001-f002:**
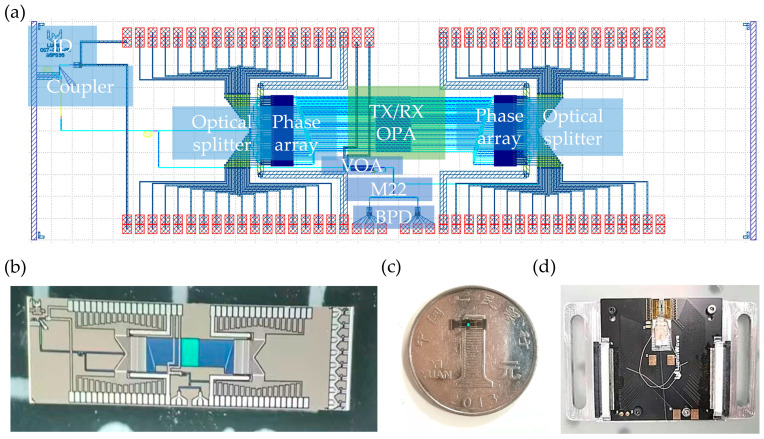
(**a**) Schematic diagram of the chip; (**b**) optical micrograph of the chip; (**c**) chip is placed on a one-yuan coin; (**d**) packaged system.

**Figure 3 micromachines-14-01001-f003:**
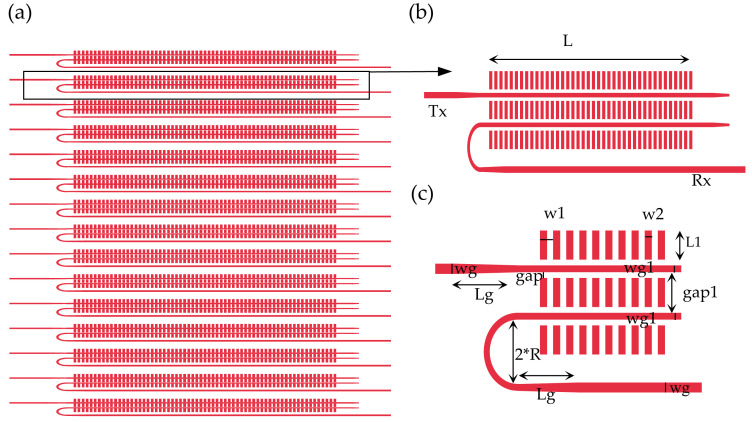
(**a**) Schematic diagram of interleaved coaxial optical phased array; (**b**) schematic diagram of interleaved coaxial antenna; (**c**) zoomed-in image of part of the antenna.

**Figure 4 micromachines-14-01001-f004:**
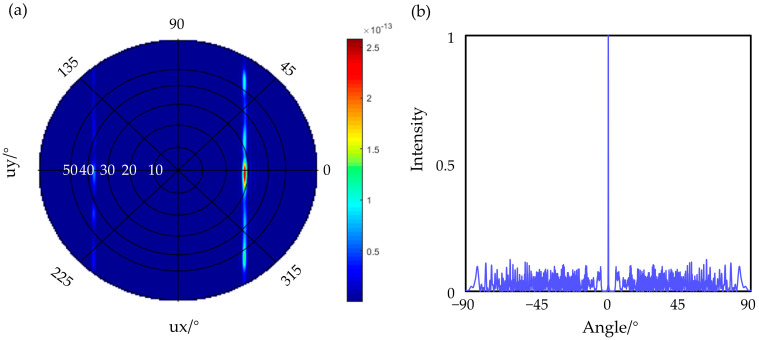
(**a**) Far-field distribution of a single antenna; (**b**) far-field distribution of an OPA with nonuniform array design.

**Figure 5 micromachines-14-01001-f005:**
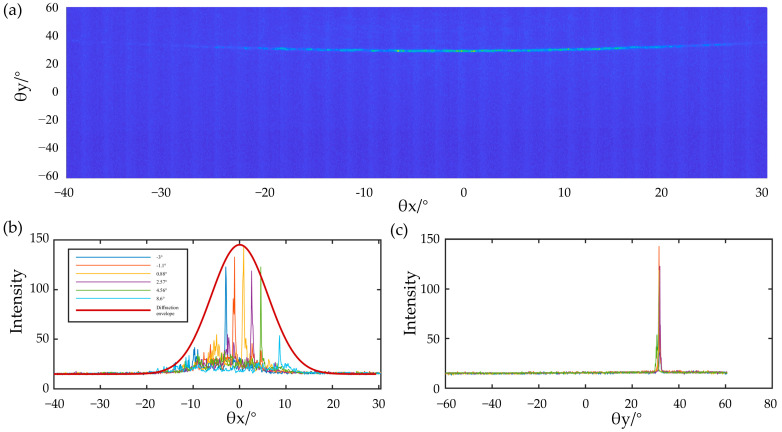
(**a**) Far-field distribution of the optical phased array when the emission optical phased array is not voltage applied to the phase shifter; (**b**) beam steering results by adjusting the phase shifters. Several scanned far-field intensity distributions in θx direction represented by different colors; (**c**) corresponding far-field distribution in θy direction.

**Figure 6 micromachines-14-01001-f006:**
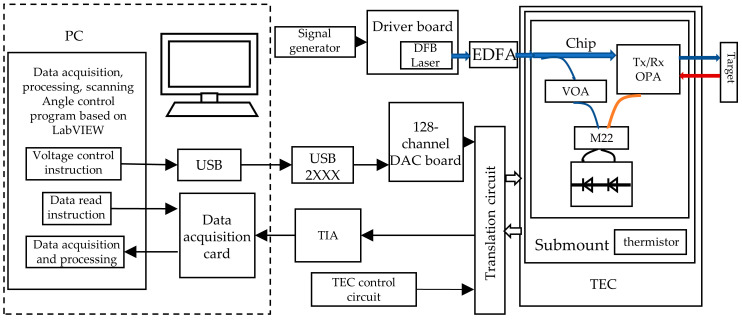
Test system for on-chip solid-state FMCW LiDAR.

**Figure 7 micromachines-14-01001-f007:**
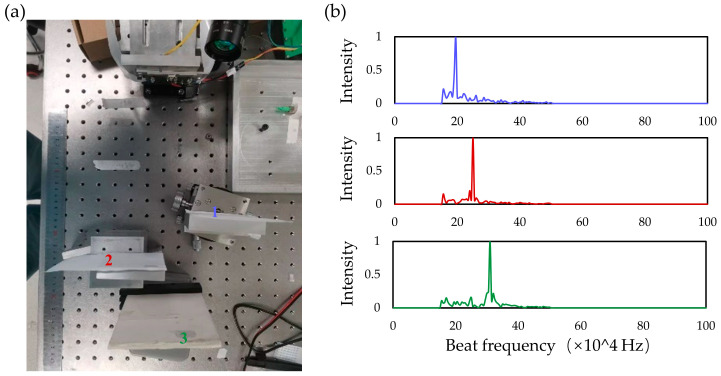
(**a**) Image of three targets placed at different angles and distances relative to the chip (**b**); FFT spectra computed from measured signal reflected from three targets scanned by OPA.

## Data Availability

Not applicable.

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
