# Peer review of "Si Photonics FMCW LiDAR Chip with Solid-State Beam Steering by Interleaved Coaxial Optical Phased Array"

_micromachines, 2023, doi:10.3390/mi14051001_

Round 1

Reviewer 1 Report

Thank you for the paper submission. Here are my comments:

  • Suggestion: Some brief outline of the alternative beam splitting based system and a comparison table showing the differences to the proposed system design would be helpful to the reader. That would also include different aspects on timing budget, scanning patterns that can be realized and power losses as mentioned in the paper but with a clear outline of breakdown for the various components. I also think that the difference should be clarified in a block diagram like figure 1.
  • In general a sketch of realizable and targeted scanning patterns would be helpful or a description of angular steps etc.
  • Line 92, the link of M12 seems to be missing in the figure.
  • Can you indicate scanning direction and beam characteristics of beam width, diffraction envelope, deflection angle, within figure 4 and 5?
  • Figure4: Is figure b supposed to show the advantage over figure a with respect to the switching from single antenna design to a nonuniform array design? Can you show the same type of far field plots for both designs?
  • Units in figures 4, 5 are missing
  • Color legends are missing in figure 5 and indication of cross section points as well
  • How are figure 5 b and c related to figure 5 a. Can that be clarified?
  • FMCW results should have some summary of the following test boundary conditions
    • Operating wavelength: Was it 1.55um as mentioned on page 4 in the grating section? Why was it chosen like that?
    • Laser power
    • Target reflecitvities
    • Chirp characteristics
    • Distance error and its main contributing sources
  • What is the impact of the optical design on the FMCW measurement?
  • I'd expect some conclusions on the actual optical design limitations from the Lidar measurement characteristics. How could those be addressed in the future? Do the sidelobes contribute to the FMCW measurement? Have you looked at thermal stability of the OPAs? What scalability constraints are existing, for example in terms of wavelength tuning to certain regimes (visible, NIR, other SWIR wavelengths) and full field scanning angles?

Author Response

Dear reviewer;

We thank the reviewer’s support and valuable comments to our manuscript. We take every comment seriously and try to addressed it. We have thoroughly considered all the comments of the reviewer and substantially revised our manuscript, and the major revised portions are marked in red. Please see the attachment for more details.

Thank you again for your help and we look forward to hearing from you.

Thank you and best regards.

Reviewer 2 Report

The authors showed an approach to design an all-solid-state FMCW on-chip lidar with an interlacing coaxial transceiver array where the energy efficiency consumption could be saved (in theory) up to 50% when compared to that of the on-chip coaxial coherent detection optical system built with 2*2 beam splitter. They demonstrated an on-chip FMCW Lidar system with 32 channels transmitter-receiver interleaved coaxial OPA with a beam width of 0.4°*0.8° and a grid-lobe rejection ratio of 6dB.

The manuscript is well-written, and the experimental/numerical results are interesting.

Here are some minor comments:

1) The authors should describe Figure 1(b). 

2) The following antenna parameter values are missed in the manuscript: gap, R, Lg, wg, and wg1.

Author Response

Dear reviewer;

We thank the reviewer’s support and valuable comments to our manuscript. All the comments and question are addressed, and the concerning revision are marked in the revised edition. Please see the attachment for more details.

Thank you again for your help and we look forward to hearing from you.

Thank you and best regards.

Reviewer 3 Report

This paper propose and demonstrate a 32-channel transmitter-receiver interleaved coaxial all-solid-state FMCW LiDAR chip. The measured beam width is 0.4°*0.8°, and the grating lobe suppression ratio is 6dB. Preliminary FMCW ranging of multiple targets scanned by OPA has been performed.

The paper claims that the system can save 50% of loss. But I disagree. The system of this paper (Figure 1a) is the same as reference [13] (see Fig. 3a). The only difference is that the two OPAs are interleaved in the grating section. Although it may seem to save grating space, but it also brings troubles. For example, larger grating spacing. Adopting non-uniform spacing design to reduce sidelobes suppressing ratio. But sidelobes suppressing ratio only 6dB. (Many paper’s results are reached over 10dB, such as [12], Photon. Res. 9(12), 2511-2518 (2021)). Some details were not introduced, such as the grating structure (Lg, gaps, W1 and W2 in Figure 3b). And the writing format is not standardized.

In summary, this paper does not show any new technologies or knowledge. I do not recommend for publication.

Author Response

Dear reviewer;

We thank the reviewer’s comments to our manuscript. We take every comment seriously and try to addressed it. Please see the attachment for more details.

Thank you again for your help and we look forward to hearing from you.

Thank you and best regards.

Round 2

Reviewer 3 Report

1, As I wrote in comments, the authors proposed Lidar system is similar to reference [14]. It not saves much place.

 2, In the reply letter, the authors claimed ‘eliminates close range blind areas’, ‘solve the alignment problem of non-coaxial systems.’ But those are all lake evidence. And this paper does not address those issues.

 3, As a comparison method, authors propose a ‘comment method’ as Fig. 1a. And proved it saves 50% of optical power. However, Fig. 1a is not a ‘comment method’. I’ve never seen it before. Unless author provides source reference. In fact, Fig. 1a is only one OPA. This paper is two OPAs.  

 4, The author claims that the structure here is coaxial. This is true, and the only innovation in this paper. However, in reference [14], two OPAs are on the same chip, and their aperture distances are also very close, which can be considered coaxial.

Author Response

Dear reviewer:

 We appreciate the reviewer’ valuable comments very much. 
